# Prevalence of malnutrition and associated factors among adult patients on antiretroviral therapy follow-up care in Jimma Medical Center, Southwest Ethiopia

**Dawit Wolde Daka** *, **Meskerem Seboka Ergiba**

Faculty of Public Health; Department of Health Policy and Management, Jimma University, Jimma, Ethiopia

* dave86520@gmail.com

## Abstract

**Data Availability Statement:** All relevant data are within the paper and its Supporting Information files.

### Background

Malnutrition especially undernutrition is the main problem that is seen over people living with HIV/AIDS and can occur at any age. Multiple factors contributed to undernutrition of HIV/AIDS patients and it need immediate identification and prompt action. The objective of this study was to assess the nutritional status of patients and identify factors associated with undernutrition among HIV/AIDS patients on follow-up care in Jimma medical center, Southwest Ethiopia.

### Methods

A cross-sectional study design was conducted from March-April 2016. Data were collected retrospectively from clinical records of HIV/AIDS patients enrolled for follow up care in ART clinic from June 2010 to January 2016. Bivariate and multivariate logistic regression analysis were performed to identify independent predictor of undernutrition.

### Results

Data of 1062 patients were included in the study. The prevalence of undernutrition (BMI<18.5 kg/m$^2$) and overweight or obesity were 34% and 9%, respectively. Out of undernourished patients, severely malnourished patients (BMI<16 kg/m$^2$) accounted of 9%. Undernutrition was more likely among widowed patients (AOR = 1.7, 95% CI, 1.03–2.79), patients with no access to water supply (AOR = 1.69, 95% CI, 1.16–2.47) and patients in the WHO clinical stage of three (AOR = 2.0, 95% CI, 1.33–2.97) and four (AOR = 3.0, 95% CI, 1.74–5.07). Moreover, the odds of undernutrition was more likely among patients with CD4 cell count of <200 cells/mm$^3$ (AOR = 2.0, 95% CI, 1.38–2.47) and patients with a functional status of bedridden (AOR = 3.6, 95% CI, 1.55–8.35) and ambulatory (AOR = 2.4, 95% CI, 1.66–3.51), respectively.

**Funding:** The research was awarded by Jimma University Mega Research Project and award was received by DWD. The funder had no role in study design, data collection and analysis, decision to publish, or preparation of the manuscript.

**Competing interests:** The authors have declared that no competing interests exist.

## Conclusion

Both undernutrition and overweight or obesity were prevalent among HIV/AIDS patients in Jimma Medical Center, Ethiopia. Undernutrition was significantly associated with clinical outcome of patients. Hence, nutritional assessment, care and support should be strengthened. Critical identification of malnourished patients and prompt interventions should be undertaken.

## Introduction

Human immune deficiency virus (HIV) and malnutrition has multifaceted and multidirectional relationships. Both are related each other in causing progressive damage to the immune system. HIV compromises nutritional status and poor nutrition further weakness the immune system of individuals, increasing susceptibility to opportunistic infections. HIV can cause or worsen malnutrition by triggering reduced food uptake, increased energy requirements and poor nutrient absorptions [1–3].

Poor nutritional status is one of the major complications of HIV and a significant factor in a full-blown AIDS. In most resource limited settings, many people who become infected with HIV may already be undernourished due to factors such as unemployment and difficulty to procure food. Food insecurity and malnutrition remained the major challenge in the progresses made to avert morbidity and mortality attributed to HIV/AIDS. Food insecurity contributes to high risk sexual behaviors, inconsistent condom use and multiple partnership. While, malnutrition leads to ART non-adherence among HIV/AIDS patients [4,5].

In Sub-Saharan Africa region, malnutrition (both undernutrition and overweight) was prevalent among HIV/AIDS patients enrolled in care [6–8]. The prevalence of adult undernutrition was 19% in Tanzania [6], 10% in Zimbabwe [7] and 19% in Senegal [9]. The prevalence of malnutrition was also high in Ethiopia; 25.2% in Butajira hospital [10] and 12.3% in Dilla university hospital [11]. The main determinants of malnutrition ranges from individual level factors to underlying factors. The high prevalence of malnutrition was determined by wealth index and educational attainment, where malnutrition decreased with increase in wealth index and educational attainment [12]. A lower CD4 cell count, sex, age, advanced HIV diseases, presence of opportunistic infections, adherence concern, inability to access food and having social support were also significantly associated with malnutrition [6,7,9,13]. Malnutrition is less likely in females and older ages (35–44 years and ≥45 years) compared to 15–24 years. Having social support and informal care giving has also reduced the odds of undernutrition [13]. On the contrary, malnutrition is more likely in those who had advanced HIV disease [6,7,9].

Despite remarkable efforts are made in increasing the treatment coverage of HIV/AIDS in the past decades, the high prevalence of HIV/AIDS and malnutrition has remained the major challenges of health systems in SSA region [14]. This was commonly attributed to the low treatment effectiveness due to factors such as non-adherence, quality and nutritional status of patients [15,16].

Ethiopia has also significantly improved the treatment coverage in the past decades and as the consequence of this promising results were observed. HIV/AIDS treatment coverage was reached to 71% in 2017. Between 2000 and 2017, new HIV infections were reduced by 90% and AIDS related mortality was reduced among adults by more than 50% [17].

However, HIV/AIDS prevalence remained high (0.9%) in Ethiopia [18] and in 2017 there were an estimated 613,000 people living with HIV/AIDS (out of which 29% were lacked access to treatments). Ethiopia is one among the 25 countries worldwide and 17 countries in Africa

region with the highest number of new HIV infections [17]. This calls for attention in improving the treatment effectiveness of existing HIV/AIDS services and one of the focus areas of this actions was strengthening the nutritional assessment, care and support services for HIV/AIDS patients. Hence, this study was aimed to estimate the prevalence of malnutrition (undernutrition and overweight or obesity) among adult HIV/AIDS patients enrolled in HIV/AIDS care in the period between 2010 and 2016 at Jimma Medical Center, a tertiary hospital in Ethiopia with an established HIV/AIDS treatment program. We also identified the predictors of undernutrition among HIV/AIDS patients.

## Materials and methods

### Study setting

Jimma Medical Center, formerly known as Jimma University Specialized Hospital, was one of the teaching hospitals in Ethiopia located in Jimma city administration, 355km Southwest direction of the capital city, Addis Ababa. It is a government hospital which is an affiliate of Jimma University providing trainings for health science students in a range of disciples. The hospital also provides a higher level of clinical care for around 15 million catchment population located in Southwest part of Ethiopia. The hospital has 36 departments, of which ART clinic was one among them. Since 2005, the hospital has been providing highly active antiretroviral therapy (HAART) for people living with HIV/AIDS (PLWHA). During the study period (July 2010 to January 2016), a total of 5554 patients were on HAART.

### Study design

Facility based cross-sectional study design was conducted. Data were collected retrospectively from clinical records of HIV/AIDS patients enrolled for follow-up care in ART clinic from July 2010 to January 2016. We retrieved data from patient cards and ART log books of adults (>15 years) from March to April 2016.

### Study participants selection

The study participants were cohort of adult HIV/AIDS patients (age >15 years old) on follow-up care. The sample size was computed using OpenEpi Version 3 sample size calculator for proportions by using assumptions of a 95% CI, the outcome factor in the population of 46.8% [19], margin of error 3% and design effect of(DEFF) of 1. The calculated sample size yields 1062.

Sample size formula (n) = [DEFF*Np (1-p)]/ [(d²/Z²$_{1-\alpha/2}$*(N-1) +p*(1-p)]; where DEFF was design effect and $N_p$ was the prevalence of outcome variable in the population (i.e., the prevalence of malnutrition).

The study participants were selected by using a systematic sampling strategy; where the clinical records of ART clients from July 2010 to January 2016 were used as a sampling frame. First, a sequential number starting from 1001 were provided to each records. The first record of a patient was selected using lottery method and every $K^{th}$ records (K = 5) were included. The value of K was obtained by dividing the total ART patients (5554) for the sample size (n = 1062).

HIV/AIDS patients who have a follow-up care in the same facility and that have key baseline information's such as demographic characteristics (age, ethnicity, and marital status) and clinical information (such as CD4 count, viral load, WHO clinical stage, weight, height, and hemoglobin level) were eligible for this study. In case when records have missed information on one of the variables, it was excluded.

## Measurement

A data collection tool, having the same structure with ART register and patient cards, was used to collect data. The tool comprised of components such as patient background information's, immunologic status, clinical and laboratory examinations (CD4 count, viral load, WHO clinical stage, weight, height) and antiretroviral regimen and opportunistic infection prophylaxis. Three clinicians who had also trained on ART service provision were recruited for data collection. Furthermore, one supervisor with background of health was recruited and monitored the overall process of data collection.

Data were collected from ART unit of the hospital in the following processes. First, data collectors approached the unit and obtained permission. Following this, patient records between July 2010 to January 2016 were retrieved from catalogs and the first patient record was selected randomly. In case when the selected patient record had incomplete information, it was dropped and the next patient record was included and reviewed. The data collection process was continued in such a way until the final sample size was attained.

## Statistical analysis

Data were checked for consistency and completeness. Then it was entered into epidata version 3.1 and exported to SPSS version 21. Descriptive statistics were conducted and variables were presented using mean, frequencies and proportion. Body Mass Index (BMI) was calculated as weight in kilograms divided by the squares of height in meters ($kg/m^2$). For the initial analysis, BMI was stratified into the WHO criteria: <18.5 (undernutrition), >18.5 to 25 (normal nutrition) and >25 $kg/m^2$ (overweight or obese). In this study both undernutrition (BMI <18.5 $kg/m^2$) and overweight or obesity (BMI>25 $kg/m^2$) were estimated [3]. Independent predictors of adult undernutrition in adult HIV/AIDS patients (such as socio-demographic and clinical characteristics) were determined by using bivariate and multivariate logistic regression analysis. In the bivariate analysis those variables with P-value < 0.05 were considered candidate variables to multivariate logistic regression model. In the multivariate analysis variables with P-value of < 0.05 were considered statistically significant and interpreted.

The covariates of the multivariate analysis were selected using enter method. While, the model fitness (goodness) was assessed by using Wald test for individual variables and maximum likelihood ratio test for the overall model.

## Ethical consideration

Research protocol was submitted to the Institute of Health of Jimma university and ethical approval was obtained (Ref no. RPGC/26/2016; March, 2016). Primarily research permission was obtained from authorities of Jimma Medical Center to access patient records for review and informed consent was not obtained from patients due to the difficulty of accessing every patients. We have accessed anonymized patient information that comprised only unique ART number of the patient and other relevant data for research. The confidentiality of all patient information was maintained through use of codes, where we have provided a sequential code to each patients starting from 1001. Paper-based data were kept in a locked cabinet and computer-based data were secured with passwords. Except the research team members, no one could access patient data. Whenever a need arises, patient data will be shared to a third party by following the legal and ethical guidelines.

## Results

Out of the total patient records (n = 1062), 357 were undernutrition patient records, 614 were normal nutrition patient records and 91 were overweight patient records. While undernutrition and normal nutrition patient records were considered in the multivariate analysis (n = 971), we reported only the prevalence of overweight patients (n = 91) ("Fig 1").

### Characteristics of patients

Table 1 presented the demographic, socio-demographic and clinical characteristics of patients on HIV/AIDS follow-care by nutritional status (undernutrition, normal nutrition and overweight or obesity).

Majority of the patients were female (61%) and in the age group of <30 years (41%). Nearly half (48%) of them were married and a bit over half of them were followers of orthodox religion (58%). While slightly over four in ten (44%) of patients have attended primary education, most (89%) were unemployed ("Table 1").

With regards to the clinical characteristics, most of patients have disclosed their HIV status to either a friend or family member (88%) and a bit over three-fourth of them have attended HIV related health education sessions previously (77%). Nearly one-thirds of patients were in the WHO clinical stage two (29%) and another 32% of patients were in the WHO clinical stage three at the start of HIV treatment (ART). Moreover, slightly over half (53%) had CD4 cell count <200 and over three-fourth (79%) of the patients had a working functional status.

The mean and median BMI of patients were 19.99 kg/m$^2$ (SD 3.41) and 19.56 kg/m$^2$, respectively. The proportion of undernutrition (BMI<18.5 kg/m$^2$), normal nutrition (BMI 18.5–24.9 kg/m$^2$) and overweight or obese (BMI≥25 kg/m$^2$) patients were 34%, 58% and 9%,

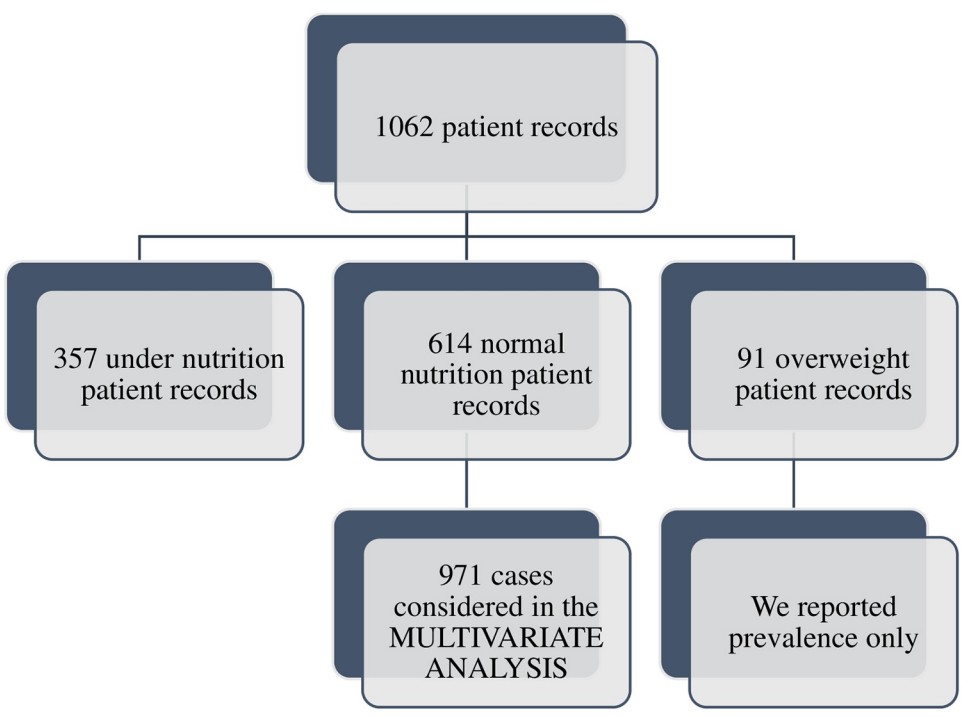

**Fig 1. Study flow diagram.**

**Table 1. Socio-demographic and clinical characteristics of patients grouped by nutritional status (BMIᵃ) in Jimma Medical Center, Southwest Ethiopia, 2016.**

| Variable | Undernutrition No (%) | Normal nutrition No (%) | Overweight or obese No (%) | Total No (%) |
|---|---|---|---|---|
| Gender | | | | |
| Male | 149(42) | 241(39) | 23(25) | 413(39) |
| Female | 208(58) | 373(61) | 68(75) | 649(61) |
| Age group (years) | | | | |
| 16-<30 | 149(42) | 253(41) | 30(33) | 432(41) |
| 30–39 | 141(39) | 227(37) | 33(36) | 401(38) |
| 40–49 | 51(14) | 100(16) | 19(21) | 170(16) |
| >=50 | 16(4) | 34(6) | 9(10) | 59(6) |
| Marital status | | | | |
| Married | 156(44) | 308(50) | 51(56) | 515(48) |
| Never married/ single | 66(18) | 98(16) | 13(14) | 177(17) |
| Separated | 52(15) | 93(15) | 5(5) | 150(14) |
| Divorced | 30(8) | 52(8) | 11(12) | 93(9) |
| Widowed | 53(15) | 63(10) | 11(12) | 127(12) |
| Religious status | | | | |
| Muslim | 120(34) | 188(31) | 34(37) | 342(32) |
| Orthodox | 196(55) | 369(60) | 47(52) | 612(58) |
| Protestant | 39(11) | 53(9) | 8(9) | 100(9) |
| Catholic | 2(1) | 4(1) | 2(2) | 8(1) |
| Educational status | | | | |
| No education | 62(17) | 126(21) | 12(13) | 200(19) |
| Primary education | 160(45) | 267(43) | 35(38) | 462(44) |
| Secondary education | 102(29) | 164(27) | 20(22) | 286(27) |
| Tertiary education | 33(9) | 57(9) | 24(26) | 114(11) |
| Employment status | | | | |
| Employed | 134(38) | 293(48) | 46(51) | 473(45) |
| Unemployed | 223(62) | 321(52) | 45(49) | 589(55) |
| Accessed to water supply | | | | |
| Yes | 285(80) | 525(86) | 76(84) | 886(83) |
| No | 72(20) | 89(14) | 15(16) | 176(17) |
| Accessed to electricity | | | | |
| Yes | 346(97) | 598(97) | 90(99) | 1034(97) |
| No | 11(3) | 16(3) | 1(1) | 28(3) |
| WHOᵇ clinical staging | | | | |
| Stage one | 58(16) | 180(29) | 44(48) | 282(27) |
| Stage two | 73(20) | 215(35) | 25(27) | 313(29) |
| Stage three | 154(43) | 172(28) | 14(15) | 340(32) |
| Stage four | 72(20) | 47(8) | 8(9) | 127(12) |
| CD4 cell count | | | | |
| <200 | 239(67) | 303(49) | 26(29) | 568(53) |
| ≥200 | 118(33) | 311(51) | 65(71) | 494(47) |
| Functional status | | | | |
| Ambulatory | 112(31) | 73(12) | 4(4) | 189(18) |
| Bed ridden | 22(6) | 9(1) | 1(1) | 32(3) |
| Working | 223(62) | 532(87) | 86(95) | 841(79) |
| Patient past opportunistic infection | | | | |

*(Continued)*

**Table 1.** (Continued)

| Variable | Undernutrition No (%) | Normal nutrition No (%) | Overweight or obese No (%) | Total No (%) |
|---|---|---|---|---|
| Yes | 297(83) | 508(83) | 68(75) | 873(82) |
| No | 60(17) | 106(17) | 23(25) | 189(18) |
| HIV/AIDS disclosure status | | | | |
| Yes | 296(83) | 549(89) | 86(95) | 931(88) |
| No | 61(17) | 65(11) | 5(5) | 131(12) |
| Accessed to spiritual care giver | | | | |
| Yes | 191(54) | 324(53) | 43(47) | 558(53) |
| No | 166(46) | 290(47) | 48(53) | 504(47) |
| Accessed to community HIV support group | | | | |
| Yes | 106(30) | 184(30) | 26(29) | 316(30) |
| No | 251(70) | 430(70) | 65(71) | 746(70) |
| Attended HIV related counseling session | | | | |
| Yes | 153(43) | 259(42) | 51(56) | 463(44) |
| No | 204(57) | 355(58) | 40(44) | 599(56) |
| Attended HIV related health education session | | | | |
| Yes | 258(72) | 486(79) | 72(79) | 816(77) |
| No | 99(28) | 128(21) | 19(21) | 246(23) |
| Diagnosed for active TB[c] | | | | |
| Yes | 63(18) | 65(11) | 4(4) | 132(12) |
| No | 294(82) | 549(89) | 87(96) | 930(88) |
| Adherence concern to ART[d] | | | | |
| Stigma | 252(71) | 435(71) | 64(70) | 751(71) |
| Afraid of medications (side effects) | 44(12) | 67(11) | 9(10) | 120(11) |
| Doubt that medications will work | 22(6) | 38(6) | 5(5) | 65(6) |
| Depressed or anxious | 27(8) | 54(9) | 9(10) | 90(8) |
| Will forget to take medications | 12(3) | 20(3) | 4(4) | 36(3) |

[a]Body Mass Index;

[b]World Health Organization;

[c]Tuberculosis;

[d]Antiretroviral Therapy

respectively. Out of undernutrition patients, severe malnutrition (BMI<16 kg/m$^2$) accounted of 9% ("Fig 2").

Undernutrition was higher in female patients (58%) and patients in the age group of <30 years old (42%). Likewise, the proportion of undernutrition was higher in married (44%), orthodox religion follower (55%), primary education level (45%) and unemployed (62%) patients ("Table 1").

Looking a closer at the distribution of undernutrition with patients' clinical characteristics, undernutrition was higher in patients with WHO clinical stage three (43%), patients with a CD4 cell count <200 cells/mm$^3$ (67%) and patients with past opportunistic infections (OI's) (62%). Whereas, patients who have disclosed their HIV status (83%) and those patients with a working functional status (62%) had a higher proportions of undernutrition.

Overweight or obesity was higher among female patients (75%) and patients in the age group of 30–39 years old (36%). Overweight or obesity was also higher in married (56%), orthodox religion follower (52%) and primary educational level (38%) patients. While nearly

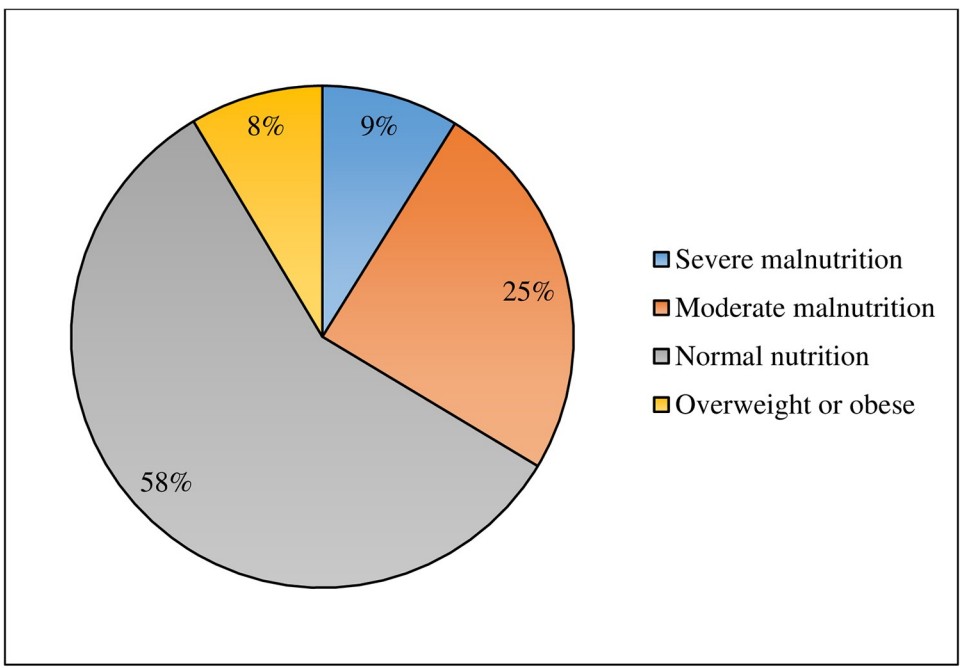

**Fig 2. Nutritional status of HIV/AIDS patients on follow-up care at Jimma Medical Center, Southwest Ethiopia, 2016.**

half of overweight or obese patients were in the WHO clinical stage one (48%), majority had a CD4 cell count $\geq$200 cells/mm$^3$ (71%) and working functional status (95%).

### Predictors of undernutrition

Table 2 presented the bivariate analysis results of undernutrition and the different independent variables. In the bivariate logistic regression analysis, marital status, employment status, access to water supply, HIV disclosure status, WHO clinical staging, CD4 cell count, functional status, attending HIV related health education sessions and being diagnosed for active TB were associated with undernutrition. These independent variables were candidates for multivariate logistic regression analysis ("Table 2").

Table 3 presented the multivariate logistic regression analysis result. In the multivariate logistic regression analysis, marital status, access to water supply, WHO clinical staging, CD4 cell count and patient functional status were identified as independent predictors of undernutrition ("Table 3").

With regards to marital status, being a widow was significantly associated with undernutrition in that undernutrition was about two times more likely among widowed patients than married patients (AOR = 1.7, 95% CI, 1.03–2.79).

The odds of undernutrition was more likely among patients who didn't access to water supply compared to those patients that did access to water supply (AOR = 1.69, 95% CI, 1.16–2.47).

Undernutrition was about two times and three times more likely among patients in the WHO clinical stage of three (AOR = 1.99, 95% CI, 1.33–2.97) and four (AOR = 2.97, 95% CI, 1.74–5.07) compared to patients in the WHO clinical stage of one, respectively.

**Table 2. Bivariate association of different variables with undernutrition among HIV/AIDS patients on follow-up care in Jimma Medical Center, Southwest Ethiopia, 2016.**

| Variable | Undernutrition n (%) | Normal nutrition n (%) | COR[a] (95% CI[b]) |
|---|---|---|---|
| Sex | | | |
| Male | 149(42) | 241(39) | Ref |
| Female | 208(58) | 373(61) | 0.90(0.69–1.18) |
| Age group (years) | | | |
| 16-<30 | 149(42) | 253(41) | Ref |
| 30–39 | 141(39) | 227(37) | 1.06(0.79–1.41) |
| 40–49 | 51(14) | 100(16) | 0.87(0.58–1.28) |
| >=50 | 16(4) | 34(6) | 0.79(0.43–1.49) |
| Marital status | | | |
| Married | 156(44) | 308(50) | Ref |
| Never married/ single | 66(18) | 98(16) | 1.33(0.92–1.92) |
| Separated | 52(15) | 93(15) | 1.10(0.75–1.63) |
| Divorced | 30(8) | 52(8) | 1.14(0.69–1.86) |
| Widowed | 53(15) | 63(10) | 1.66(1.09–2.51)* |
| Religious status | | | |
| Muslim | 120(34) | 188(31) | Ref |
| Orthodox | 196(55) | 369(60) | 0.83(0.62–1.11) |
| Protestant | 39(11) | 53(9) | 1.15(0.72–1.85) |
| Catholic | 2(1) | 4(1) | 0.78(0.14–4.34) |
| Educational status | | | |
| No education | 62(17) | 126(21) | Ref |
| Primary education | 160(45) | 267(43) | 1.22(0.85–1.75) |
| Secondary education | 102(29) | 164(27) | 1.26(0.85–1.87) |
| Tertiary education | 33(9) | 57(9) | 1.18(0.69–1.99) |
| Employment status | | | |
| Employed | 134(38) | 293(48) | Ref |
| Unemployed | 223(62) | 321(52) | 1.52(1.16–1.98)* |
| Accessed to water supply | | | |
| Yes | 285(80) | 525(86) | Ref |
| No | 72(20) | 89(14) | 1.49(1.06–2.09)* |
| Accessed to electricity | | | |
| Yes | 346(97) | 598(97) | Ref |
| No | 11(3) | 16(3) | 1.19(0.55–2.59) |
| WHO[c] clinical staging | | | |
| Stage one | 58(16) | 180(29) | Ref |
| Stage two | 73(20) | 215(35) | 1.05(0.71–1.57) |
| Stage three | 154(43) | 172(28) | 2.78(1.93–4.01)* |
| Stage four | 72(20) | 47(8) | 4.75(2.97–7.62)* |
| CD4 cell count (cells/mm$^3$) | | | |
| <200 | 239(67) | 303(49) | 2.08(1.59–2.73)* |
| ≥200 | 118(33) | 311(51) | Ref |
| Functional status | | | |
| Ambulatory | 112(31) | 73(12) | 3.66(2.62–5.11)* |
| Bed ridden | 22(6) | 9(1) | 5.83(2.64–12.86)* |
| Working | 223(62) | 532(87) | Ref |
| Patients' past opportunistic infections | | | |

*(Continued)*

**Table 2.** (Continued)

| Variable | Undernutrition n (%) | Normal nutrition n (%) | COR[a] (95% CI[b]) |
|---|---|---|---|
| Yes | 297(83) | 508(83) | 1.03(0.73–1.46) |
| No | 60(17) | 106(17) | Ref |
| HIV/AIDS disclosure status | | | |
| Yes | 296(83) | 549(89) | 0.58(0.39–0.84)* |
| No | 61(17) | 65(11) | Ref |
| Accessed to spiritual care giver | | | |
| Yes | 191(54) | 324(53) | 1.03(0.79–1.34) |
| No | 166(46) | 290(47) | Ref |
| Accessed to community HIV support group | | | |
| Yes | 106(30) | 184(30) | 1.01(0.76–1.35) |
| No | 251(70) | 430(70) | Ref |
| Attended HIV related counseling session | | | |
| Yes | 153(43) | 259(42) | 1.03(0.79–1.34) |
| No | 204(57) | 355(58) | Ref |
| Attended HIV related health education session | | | |
| Yes | 258(72) | 486(79) | 0.69(0.51–0.93)* |
| No | 99(28) | 128(21) | Ref |
| Diagnosed for active TB[d] | | | |
| Yes | 63(18) | 65(11) | 1.81(1.25–2.63)* |
| No | 294(82) | 549(89) | Ref |
| Adherence concerns to ART[e] | | | |
| Stigma | 252(71) | 435(71) | Ref |
| Afraid of medications (side effects) | 44(12) | 67(11) | 1.13(0.75–1.71) |
| Doubt that medications will work | 22(6) | 38(6) | 0.99(0.58–1.73) |
| Depressed or anxious | 27(8) | 54(9) | 0.86(0.53–1.41) |
| Will forget to take medications | 12(3) | 20(3) | 1.04(0.49–2.15) |

[a]Crude (unadjusted) odds ratio;

[b]Confidence interval;

[c]World Health Organization;

[d]Tuberculosis;

[e]Antiretroviral Therapy

*P-value<0.05

HIV/AIDS patients with CD4 cell count <200 cells/mm$^3$ were two times more likely under nourished than patients with CD4 cell count ≥200 cells/mm$^3$ (AOR = 1.85, 95% CI, 1.38–2.47).

Moreover, the odds of undernutrition was more likely among patients with ambulatory (AOR = 2.41, 95% CI, 1.66–3.51) and bedridden (AOR = 3.59, 95% CI, 1.55–8.35) functional status than those patients with working functional status, respectively.

In the bivariate logistic regression analysis, employment status, HIV disclosure status, being diagnosed for active TB and attending HIV related health education sessions were significantly associated with undernutrition. However, the associations of these variables were not maintained after adjusting for all independent variables.

**Table 3. Multivariate association of different variables with undernutrition among HIV/AIDS patients on follow-up care in Jimma Medical Center, Southwest Ethiopia, 2016.**

| Variable | Undernutrition No (%) | Normal nutrition No (%) | COR[a] (95% CI[b]) | AOR[c](95% CI) |
|---|---|---|---|---|
| Marital status | | | | |
| Married | 156(44) | 308(50) | Ref | Ref |
| Never married/ single | 66(18) | 98(16) | 1.33(0.92–1.92) | 1.31(0.87–1.96) |
| Separated | 52(15) | 93(15) | 1.10(0.75–1.63) | 0.96(0.62–1.48) |
| Divorced | 30(8) | 52(8) | 1.14(0.69–1.86) | 1.11 (0.65–1.91) |
| Widowed | 53(15) | 63(10) | 1.66 (1.09–2.51)* | 1.7(1.03–2.79)* |
| Employment status | | | | |
| Employed | 134(38) | 293(48) | Ref | Ref |
| Unemployed | 223(62) | 321(52) | 1.52(1.16–1.98)* | 1.29(0.96–1.74) |
| Accessed to water supply | | | | |
| Yes | 285(80) | 525(86) | Ref | Ref |
| No | 72(20) | 89(14) | 1.49(1.06–2.09)* | 1.69(1.16–2.47)* |
| WHO[d] clinical staging | | | | |
| Stage one | 58(16) | 180(29) | Ref | Ref |
| Stage two | 73(20) | 215(35) | 1.05(0.71–1.57) | 0.92(0.61–1.39) |
| Stage three | 154(43) | 172(28) | 2.78(1.93–4.01)* | 1.99(1.33–2.97)* |
| Stage four | 72(20) | 47(8) | 4.75(2.97–7.62)* | 2.97(1.74–5.07)* |
| CD4 cell count cells/mm$^3$ | | | | |
| <200 | 239(67) | 303(49) | 2.08(1.59–2.73)* | 1.85(1.38–2.47)* |
| ≥200 | 118(33) | 311(51) | Ref | Ref |
| Functional status | | | | |
| Ambulatory | 112(31) | 73(12) | 3.66(2.62–5.11)* | 2.41(1.66–3.51)* |
| Bed ridden | 22(6) | 9(1) | 5.83(2.64–12.86)* | 3.59(1.55–8.35)* |
| Working | 223(62) | 532(87) | Ref | Ref |
| HIV disclosure status | | | | |
| Yes | 296(83) | 549(89) | 0.58(0.39–0.84)* | 0.75(0.49–1.15) |
| No | 61(17) | 65(11) | Ref | Ref |
| Attended HIV related health education session | | | | |
| Yes | 258(72) | 486(79) | 0.69(0.51–0.93)* | 0.80 (0.57–1.13) |
| No | 99(28) | 128(21) | Ref | Ref |
| Diagnosed for active TB[e] | | | | |
| Yes | 63(18) | 65(11) | 1.81(1.25–2.63)* | 0.83(0.53–1.29) |
| No | 294(82) | 549(89) | Ref | Ref |

[a]Crude (unadjusted) odds ratio;

[b]Confidence interval;

[c]Adjusted odds ratio;

[d]World Health Organization;

[e]Tuberculosis

*P-value<0.05

## Discussion

This study was primarily intended to estimate the prevalence of undernutrition and identify the potential predictors of undernutrition among adult HIV/AIDS patients who were enrolled on care or who were on ART treatments. It was also aimed to estimate the prevalence of overweight or obesity among adult HIV/AIDS patients.

The study have indicated that the overall prevalence of undernutrition and overweight or obesity were 34% and 9%, respectively. Out of undernutrition, severe malnutrition (BMI<16 kg/m$^2$) accounted of 9%. We found that patients marital status, access to water supply, WHO clinical staging, CD4 cell count and functional status were significantly associated with undernutrition.

The prevalence of undernutrition was higher compared to studies conducted in different parts of Ethiopia; 12.3% in Dilla university hospital [11], 25.2% in Butajira hospital [10], 18.2% in Arba Minch area public health facilities [20], 30% in East Hararghe zone hospitals [21] and 27% in Nekemte referral hospital [22]. Similarly, the prevalence is also much higher compared to studies done in different parts of the world; 19.5% in Tanzania [6], 10% in Zimbabwe [7] and 19.2% in Senegal [9]. The difference in prevalence of malnutrition might be due to difference in socio-economic and other factors that may predispose the community to problem, such as food habit and culture. However, the prevalence of severe malnutrition (BMI<16kg/m$^2$) was comparable to studies done at Tanzania (9%) [6] and Butajira hospital(9%) [10].

In our study undernutrition was higher among female patients than male. This is comparable to other studies done in different parts of Ethiopia; Southern [11] and Eastern Ethiopia [21]. Yet different from studies done in SSA countries [6,7] and slightly higher among male patients in study done in Butajira hospital, Southern Ethiopia [10]. This implies that undernutrition is more common among females and this might be related to the socio-economic status of women, access to information and other predisposing factors that may affect food intake of women in the community [23]. In Ethiopia, most women were already undernourished than men in the general population. For instance, 24% of women aged 15–49 years were anemic and 22% were thin (BMI<18.5 kg/m$^2$) [18]. Further, lower proportion of female were attended school compared to male and the employment status was much higher among men than women. Similarly, most women were not accessed to mass media and internet compared to men [18].

Our study also revealed that, undernutrition was higher in younger, married, primary education level and unemployed patients compared to their counter parts. As the age of patient's increased, the prevalence of undernutrition was decreased in the study area. This is comparable with studies done in different parts of the world [6,7] and Ethiopia [10,21], where undernutrition is more prevalent among the younger adults and overweight is more prevalent among older patients. Additionally, studies also supports the higher proportion of undernutrition among unemployed patients and patients in the lower educational status [11,24,25]. While studies done in Northern and Southern Ethiopia indicated contradicting finding in which undernutrition was higher in older age [11,26]. Undernutrition was also much higher in patients in WHO clinical stage three and four, CD4 cell count less than 200 cells/mm$^3$ and in patients with a past opportunistic infections. This indicates interlink between HIV infection and poor nutritional status. HIV compromises the nutritional status and poor nutrition further weakens the immune system of individuals, increasing susceptibility to opportunistic infections [2,23,27,28].

In the multivariate logistic regression, patient marital status, access to water supply, WHO clinical staging, CD4 cell count and functional status were statistically associated with undernutrition in the study area. In the binary logistic regression variables such as employment status, being diagnosed for active TB, attending HIV related health education sessions and HIV disclosure status of patients were significantly associated with undernutrition. Though, these variables didn't show a statistical significant associations while adjusted for other variables.

Our study revealed that, being a widow and poor access to water supply were significantly associated with undernutrition among HIV/AIDS patients. The association between marital status and undernutrition could might be due to the relationship of marital status and household headship and other social and economic status of individuals [23,29]. It might be also

related to psychological conditions of individuals, where married patients better follow HIV treatments as they get required supports from their partners. On the other hand, the association of undernutrition and access to water supply is due to the fact that poor sanitation was related to diseases in patients and disease in itself leads to poor nutritional status of individuals [30,31].

HIV/AIDS patients who were in the WHO clinical stage of three and four were two times and three times more likely undernourished compared to those in the WHO clinical stage of one, respectively. Additionally, the odds of undernutrition was more likely among patients with CD4 cell count of <200 cells/mm$^3$ and patients with functional status of bedridden and ambulatory. Undernutrition was nearly four times and two times more likely among bedridden and ambulatory patients compared to patients who have a working functional status, respectively. Our findings were supported by other studies done in other settings else [6,7,10,11,13,20,21,24–26].

Undernutrition was the major factor in ensuring treatment effectiveness and thus, nutritional assessments, care and support for HIV/AIDS patients should be strengthened. Poor adherence to HIV treatments or drug is caused by lack of access to food or food insecurity. On the other hand, poor treatment adherence leads to infections and suppressed immunity of patients [9]. Mortality is more likely among immune compromised HIV/AIDS patients and supplementation of therapeutic food can improve the nutritional status of patients [32].

Our study have assessed the prevalence of malnutrition (undernutrition and overweight or obesity) among patients who were enrolled in care in the period between 2010 and 2016. Sufficient sample size was used to estimate the prevalence of both undernutrition and overweight or obesity. Yet, our study has the following limitations. The study was based on secondary data (chart review) and thus, may not address all of the variables that may affect undernutrition and may be subject to incomplete data bias.

## Conclusion

This study showed that the prevalence of malnutrition (both undernutrition and overweight) was high compared to other settings in Ethiopia. It was also revealed that WHO clinical AIDS staging three and four, CD4 cell count <200 cells/mm$^3$, functional status, patient marital status and access to water supply were significantly associated with undernutrition among adult HIV/AIDS patients. This calls the government to give due attention to strengthening the HIV/AIDS treatment, care and support services at hospitals. HIV/AIDS treatment services should be supported with nutritional assessment, supplementation, counseling, care and support to patients. A comprehensive nutritional assessment and support should be provided for all patients who are on a follow-up care. Moreover, community support to patients should be strengthened, as social determinants of health may also interact with the effectiveness HIV/AIDS treatments.

## Supporting information

**S1 File. Minimum data set.**
(XLSX)

## Acknowledgments

The authors would like to thank Jimma Medical Center administration and Workers at Antiretroviral Therapy (ART) unit for cooperating with and supporting the research work.

## Author Contributions

**Conceptualization:** Dawit Wolde Daka.

**Data curation:** Dawit Wolde Daka.

**Formal analysis:** Dawit Wolde Daka.

**Funding acquisition:** Dawit Wolde Daka.

**Investigation:** Dawit Wolde Daka.

**Methodology:** Dawit Wolde Daka.

**Project administration:** Dawit Wolde Daka.

**Supervision:** Dawit Wolde Daka.

**Validation:** Dawit Wolde Daka.

**Visualization:** Dawit Wolde Daka.

**Writing – original draft:** Dawit Wolde Daka.

**Writing – review & editing:** Dawit Wolde Daka.

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
