## [Decision Letter · Decision Letter 0]

13 Dec 2019

PONE-D-19-30011

Prevalence of malnutrition and associated factors among adult patients on Antiretroviral Therapy follow up care in Jimma Medical Center, Southwest Ethiopia.

PLOS ONE

Dear Mr. Daka,

Thank you for submitting your manuscript to PLOS ONE. After careful consideration, we feel that it has merit but does not fully meet PLOS ONE’s publication criteria as it currently stands. Therefore, we invite you to submit a revised version of the manuscript that addresses the points raised during the review process.

Be sure to:

We recommend you pay specific attention to major comments from reviewer 1, and address them 

We would appreciate receiving your revised manuscript by Jan 27 2020 11:59PM. To enhance the reproducibility of your results, we recommend that if applicable you deposit your laboratory protocols in protocols.io, where a protocol can be assigned its own identifier (DOI) such that it can be cited independently in the future. For instructions see: http://journals.plos.org/plosone/s/submission-guidelines#loc-laboratory-protocols

We look forward to receiving your revised manuscript.

Kind regards,

Simeon-Pierre Choukem

Academic Editor

PLOS ONE

Journal Requirements:

2. In ethics statement in the manuscript and in the online submission form, please provide additional information about the patient records/samples used in your retrospective study. Specifically, please ensure that you have discussed whether all data/samples were fully anonymized before you accessed them and/or whether the IRB or ethics committee waived the requirement for informed consent. If patients provided informed written consent to have data/samples from their medical records used in research, please include this information.

3. Please include your tables as part of your main manuscript and remove the individual files. Please note that supplementary tables (should remain/ be uploaded) as separate "supporting information" files.

Reviewers' comments:

Reviewer's Responses to Questions

**Comments to the Author**

1. Is the manuscript technically sound, and do the data support the conclusions?

Reviewer #1: Yes

Reviewer #2: Yes

2. Has the statistical analysis been performed appropriately and rigorously? 

Reviewer #1: Yes

Reviewer #2: Yes

3. Have the authors made all data underlying the findings in their manuscript fully available?

Reviewer #1: Yes

Reviewer #2: Yes

4. Is the manuscript presented in an intelligible fashion and written in standard English?

Reviewer #1: No

Reviewer #2: No

5. Review Comments to the Author

Reviewer #1: General

The article address a major issue among HIV infected patients, especially in Africa and other low income settings. The study hypothesis and objectives are well stated. The authors have evaluated the prevalence and risk factors of under nutrition among patients receiving ART at a university hospital in Ethiopia. They found a high (36%) prevalence of under nutrition ; associated with marital status (widow), advance WHO clinical stage (III-IV), and low CD4 count (<200 cell/ul). While this topic has already being studied in several settings, local data are crucial to design local actions.

Major comments

The reference to malnutrition is sometimes confusing as it include both undernutrition and overweight. Reader may feel more comfortable if authors focus on undernutrition (the main topic of the study). This could be stated already in the title.

Authors state malnutrition as The main problem among people living with HIV. Maybe this statement could be attenuated, indicating food security and under nutrition as a major threat to HIV programs.

It would have been interesting to have information about diet, food security and incomes (eg. Wealth index) as these factors are known to affect nutritional status. Advance HIV status and low cd4 count found in this study, as well as under nutrition, may all be surrogates of poverty or deficient diet.

With the high prevalence (55.8%) of patients with less than 200 CD4, it would be interesting to indicate the prevalence of patients in treatment failure. The observed severe under nutrition could be a wasting syndrome related to a treatment failure and disease progression.

The study design is stated as cross-sectional. Readers may have difficulties to understand the patients inclusion process. It may be useful to clarify this section, especially the used "lottery method" (random selection ?) and the "Kth records identified for abstraction", lines 111-113.

The authors seem to make a confusion between prevalence in a study group and weight of the group within the whole population. Thus, most of the undernourished being married (table 3) does not necessarily mean that undernutrition is most prevalent among married as stated at line 173. This mistake apply to other variables mentioned in the paragraph "Prevalence of malnutrition". Conversely, in paragraph addressing factors associated with malnutrition, authors should use "proportion" instead of "number" (line 186 and beyond).

Table 1,2 and 3 can be merged to keep only two tables, with 3 results columns each: 1- Normal nutritional status, 2- Under nutrition and 3- total

To ensure easy reading of the paper, English language should be improved.

Minor comments

Details about countries prevalence and the risks factors of malnutrition discussed in the introduction (lines 58-60 and 63-69) would be more useful in the discussion

Line 151: Readers may be interested to know why 9% of the records were not completed (missing data?)

Line 54: Food insecurity (and not malnutrition by itself) increases high risk sexual behaviors

Line 72: author may state the overweigh is also "a major" problem and not "the main"

Line 121: "data abstraction tool" here probably refer to a "data collection tool"

Line 142: the section about patients confidentiality may be simplified.

Reviewer #2: No research has been done to this scope, wide range of duration and large sample size in Ethiopia. Because of limited number of researches are there to establish volume of evidences for different contradicting findings regarding the factors, it contributes to conduct valid systematic review and Meta analysis. The authors should revisit the method section so that it will be clear for readers. Write population, inclusion and exclusion criteria and the analysis section clearly. Let the language experts edit the manuscript. Discussion should address some plausible scientific explanation in addition to comparison. Finally if these comments are refined, I wish it will be accepted publication.

6. PLOS authors have the option to publish the peer review history of their article (what does this mean?). If published, this will include your full peer review and any attached files.

Reviewer #1: No

Reviewer #2: No

---

## [Author Response · Author response to Decision Letter 0]

22 Jan 2020

We appreciate the comments provided by the editor and reviewers on the original research article submitted to PLOS ONE. In this letter we would like to provide our responses both for the comments provided by the editor and the reviewers separately.

Response to the Editor:

A per the comments provided by the Editor, we have modified the formats of the manuscript (the Title page and tables/figures) to be compatible with the PLOS ONE guidelines. 

Regarding to the ethics statement, all patient’s data were initially anonymized in that we have accessed all patient records with only their unique ART number. Though, as this unique ART number might let identification of patients; patient data were collected by using unique codes for each records starting from 1001. As data were collected from secondary sources (patient records), we were unable to take informed consent from each patients. Instead, we have obtained a permission from authorities of the Medical Center based on the approval letter received from the Ethical Review Committee. The requirements of informed consent was waived by the IRB or ethics committee. 

With regards to data sharing, we have included the de-identified or anonymized patient data into the supplementary information’s. The format of data included in the supplementary information’s was excel or spreadsheets. 

Finally, the corresponding author have created ORCID iD and it was validated by Editorial manager. 

Response to the reviewers:

Response to reviewer 1:

The main aim of this article was to estimate the prevalence of undernutrition and identify the predictors of undernutrition in adult HIV/AIDS patients on follow-up care. We have also estimated the prevalence of overweight or obesity among HIV/AIDS patients. This is the reason why we have defined the title as the prevalence of malnutrition instead of undernutrition and we repeatedly utilized the term malnutrition in the manuscript. 

We have addressed the detail description of the study methods (study design, study population and participant’s selection processes) in the methods and materials section of the current or revised manuscript. In addition we have also clearly presented the study participation in the results section of the manuscript with support of the study flow diagram (Fig 1). Out of the total study participants (n=1062), normal nutrition, under nutrition and overweight or obesity accounted of 614, 357 and 91; respectively. The missing data (9%) in the previous version of the manuscript was observed as we didn’t clearly indicate the denominator in estimating overweight or obesity (lack of clarity in data presentation). Overall, in this version of the manuscript; we tried to adequately address the comments provided in the methods and results section by the reviewer. 

As we have used secondary data for this specific article and only limited variables were included, we didn’t assess the food insecurity and wealth index of patients. Hence, the variables addressed were limited to the one presented in this article/manuscript. 

We have addressed the comments provided in the results section. We have merged table 1, 2 and 3. Hence, the total number of tables in the manuscript was reduced to 3. We have also incorporated the comments provided in the ‘Prevalence of malnutrition’ sub-heading of the Results section of the manuscript. Additionally, we have also addressed the minor comments provided by the reviewer. 

Response to reviewer 2: 

Most of the comments provided by the reviewer are related to the clarity of the method section and language. In the revised version of the manuscript, we have clearly presented the study methods (including the study design, study population, sampling, inclusion/exclusion criteria and analysis). We have also revised the language. 

Kindly regards,

---

## [Decision Letter · Decision Letter 1]

19 Feb 2020

Prevalence of malnutrition and associated factors among adult patients on Antiretroviral Therapy follow-up care in Jimma Medical Center, Southwest Ethiopia.

PONE-D-19-30011R1

Dear Dr. Daka,

We are pleased to inform you that your manuscript has been judged scientifically suitable for publication and will be formally accepted for publication once it complies with all outstanding technical requirements.

With kind regards,

Simeon-Pierre Choukem

Academic Editor

PLOS ONE

Additional Editor Comments (optional):

Reviewers' comments:

Reviewer's Responses to Questions

**Comments to the Author**

1. If the authors have adequately addressed your comments raised in a previous round of review and you feel that this manuscript is now acceptable for publication, you may indicate that here to bypass the “Comments to the Author” section, enter your conflict of interest statement in the “Confidential to Editor” section, and submit your "Accept" recommendation.

Reviewer #1: All comments have been addressed

Reviewer #2: All comments have been addressed

2. Is the manuscript technically sound, and do the data support the conclusions?

Reviewer #1: Yes

Reviewer #2: Yes

3. Has the statistical analysis been performed appropriately and rigorously? 

Reviewer #1: Yes

Reviewer #2: Yes

4. Have the authors made all data underlying the findings in their manuscript fully available?

Reviewer #1: Yes

Reviewer #2: Yes

5. Is the manuscript presented in an intelligible fashion and written in standard English?

Reviewer #1: Yes

Reviewer #2: Yes

6. Review Comments to the Author

Reviewer #1: (No Response)

Reviewer #2: (No Response)

7. PLOS authors have the option to publish the peer review history of their article (what does this mean?). If published, this will include your full peer review and any attached files.

Reviewer #1: No

Reviewer #2: No

---

## [Editor Report · Acceptance letter]

25 Feb 2020

PONE-D-19-30011R1 

Prevalence of malnutrition and associated factors among adult patients on Antiretroviral Therapy follow-up care in Jimma Medical Center, Southwest Ethiopia. 

Dear Dr. Daka:

I am pleased to inform you that your manuscript has been deemed suitable for publication in PLOS ONE. Congratulations! Your manuscript is now with our production department. 

With kind regards,

on behalf of

Dr. Simeon-Pierre Choukem 

Academic Editor

PLOS ONE